# Proposing an Optimal Occlusal Angle for Minimizing Masticatory and Cervical Muscle Activity in the Supine Position: A Resting EMG and Mixed-Effects Modeling Study

**DOI:** 10.3390/medicina61071274

**Published:** 2025-07-15

**Authors:** Kyung-Hee Kim, Chang-Hyung Lee, Sungchul Huh, Byong-Sop Song, Hye-Min Ju, Sung-Hee Jeong, Yong-Woo Ahn, Soo-Min Ok

**Affiliations:** 1Department of Oral Medicine, Busan Paik Hospital, Inje University, Pusan 47392, Republic of Korea; aussiedent@hotmail.com; 2Rehabilitation Medicine, Pusan National University Yangsan Hospital, Yangsan 50612, Republic of Korea; aarondoctor@gmail.com (C.-H.L.); dr.huhsc@gmail.com (S.H.); 3Rehabilitation Medicine, Research Institute for the Convergence of Biomedical Science and Technology, Pusan National University School of Medicine, Yangsan 50612, Republic of Korea; 4School of Life Sciences, Gwangju Institute of Science and Technology, 123 Cheomdan-gwagiro, Gwangju 61005, Republic of Korea; bssongstat@gmail.com; 5Department of Oral Medicine, Dental and Life Science Institute, Pusan National University, Yangsan 50612, Republic of Korea; jc2wma@pusan.ac.kr (H.-M.J.); drcookie@pusan.ac.kr (S.-H.J.); ahnyongw@pusan.ac.kr (Y.-W.A.); 6Dental Research Institute, Pusan National University Dental Hospital, Yangsan 50612, Republic of Korea

**Keywords:** occlusal angle, temporomandibular disorders, muscle fatigue, surface electromyography, masticatory muscle, cervical muscle, pillow height

## Abstract

*Background and Objectives*: The occlusal angle (OA), influenced by pillow height, may affect muscle tension in the head and neck. However, its optimal range for minimizing muscle activation has not been clearly defined. This study aimed to investigate the effects of OA on the resting muscle activity of masticatory and cervical muscles and to identify an optimal OA range using cluster analysis and linear mixed-effects modeling. *Materials and Methods*: The resting muscle activities of the masseter (MAS), temporalis (TEM), sternocleidomastoid (SCM), and posterior vertebral muscles (PVM) were measured at OA conditions modulated by pillow heights of 0, 5, and 10 cm at 0, 1, and 5 min in the supine position. Intraclass correlation coefficients (ICCs) assessed measurement reliability. Statistical analyses included ANOVA, ROC curve analysis, k-means clustering, and linear mixed-effects models. *Results*: MAS and TEM resting muscle activity ratio (RMR) significantly increased with larger OA values (*p* < 0.001), while SCM showed decreased activation (*p* = 0.001). An OA range of 105°–111° was identified as the center of a low-activity cluster, and an upper cut-off of 138° was associated with potential muscular overload. ICC values for MAS and SCM ranged from 0.82 to 0.89, indicating excellent test–retest reliability. *Conclusions*: OA modulated by pillow height is a modifiable factor that influences muscle activity. An OA of 105°–111° may serve as a practical comfort zone, especially for individuals at risk of TMDs.

## 1. Introduction

Adequate occlusal angle (OA) modulated by pillow height is known to be crucial to decrease muscle pain. Neck alignment can be greatly affected by the shape and height of the pillow [1]. Prolonged use of an uncomfortable pillow height and shape can lead to neck strain, headaches, and chronic cervical spine disease [2,3]. An optimal pillow size and shape is necessary for the sleeper to be comfortable and for the associated muscles to relax. Some researchers have studied how to most appropriately modify the pillow configuration without creating undesired muscle activation [4,5]. Among these studies, most investigated the association between neck and shoulder muscle activities and pillow configuration. Previous research has demonstrated that pillow height significantly affects cervical muscle activity and comfort, with a 10 cm height showing the lowest EMG activity and highest comfort level among 5, 10, and 14 cm pillows [6]. A recent systematic review emphasized that both excessively high and low pillows can disrupt cervical spine curvature and increase muscle stress [7]. These findings highlight the importance of optimized pillow height in minimizing neuromuscular load, which is particularly relevant for individuals with temporomandibular disorders (TMDs). However, studies on the association between the activity of the jaw muscles and the OA modulated by pillow height and shape are rare [4,5,6,7,8,9]. Furthermore, an optimal OA has yet to be elucidated.

Most of these previous investigations failed to study the muscle activities according to OA modulated by pillow height by classifying human weight and height [4,5,6]. However, one study did find that an increase in OA was not directly correlated with an increase in the cervicohorizontal angle (also called the neck angle) [10]. The cervicohorizontal angles may differ among individuals with varying heights and weights at the same pillow height (Figure 1A), and an individual may lie on the same pillow at different positions so that the craniocervical angles vary (Figure 1B) [11]. Thus, it may be difficult to control the cervicohorizontal and craniocervical angles by simply adjusting pillow height. Therefore, considering the OA rather than the pillow height alone may better reflect head and neck alignment, which can influence the levels of muscle activity.

Since the masticatory muscles have different attachment sites from the back and neck muscles, the masseter and temporalis, which are important masticatory muscles, are not attached to the neck and back, but are instead attached to the skull and mandible; thus, the cervicohorizontal angle may not reflect posture changes which can affect masticatory muscle activities. Therefore, it was hypothesized that the OA was the angle of the maxillary occlusal plane with respect to the ground (Figure 1C), and it would be more suitable to reflect the masticatory muscle activities than the cervicohorizontal angle. Moreover, the craniocervical angle (CCA), often measured from external landmarks such as the vertex and menton, may vary depending on slight mouth opening, which frequently occurs during pillow-supported supine rest. Since the extent of mandibular opening can change with pillow height, the CCA may not reliably represent the actual occlusal posture. In contrast, the occlusal angle (OA), defined by the maxillary occlusal plane and ground, remains consistent irrespective of small mandibular shifts. This angle therefore offers a more stable and functionally relevant parameter to evaluate masticatory muscle behavior in the resting supine position. The biomechanical rationale for considering the OA is supported by previous finite element modeling studies, which have demonstrated that mandibular posture directly influences load distribution and muscle activation in the masticatory system [12]. Additionally, changes in head posture can alter occlusal contact points and masticatory muscle alignment, as demonstrated using occlusal force-mapping systems such as T-Scan [13]. The OA and muscle activities were measured with the participants in the supine position, as sleep duration in the supine position is much longer than that in the left or right lateral position (Figure 1D) [14].

On this basis, the purpose of our research was to analyze the changes in masticatory and cervical muscle (MCM) activities according to OA changes and to suggest the most suitable OA modulated by pillow height for an individual.

## 2. Materials and Methods

### 2.1. Design and Participants

This was a repeated-measures experimental study designed to assess changes in masticatory and cervical muscle activities under different OA conditions modulated by pillow height. The sample size was calculated with 90% power and 30% drop rate. A total of 23 healthy adults (9 men, 14 women; mean age, 42.44 ± 15.36) were recruited between August and October 2015 from among 30 volunteers of both sexes and various ages, through local advertisements and word of mouth. To reduce confounding muscle activity variation, seven participants with temporomandibular disorder (TMD), headaches, or other medically diagnosed conditions were excluded following dental evaluation. Seven participants with temporomandibular disorder (TMD)-associated pain, headaches, or any illness diagnosed by a dentist were excluded. In addition, all participants had normal, healthy dentition and no severe skeletal disorders. The participant flow through each stage of the study is illustrated in Appendix A. All participants voluntarily provided informed written consent. This study was approved by the Institutional Review Board of the Hospital (no. PNUH-042015014 and date of approval 19 August 2015).

### 2.2. Pillow Height

We tested the patients using a rectangular box-shaped pillow (a high pillow [10 cm] or a medium pillow [5 cm]) or in the no-pillow condition (head resting on the ground with neck support). The selection of the height of 5 cm was due to its reported suitability for normal pulmonary function [15], and 10 cm was selected because it was reported to decrease the neck and mid-upper back muscle fatigue [16]. All participants were exposed to all three pillow height conditions (0, 5, and 10 cm) in a fixed sequence rather than a randomized order, to minimize inter-subject variability and avoid potential learning or fatigue bias. The same testing order was applied consistently across participants. Neck support was provided to all participants to maintain their own cervical curvature and reduce unnecessary muscle activity. All pillows had a density of 2 g/cm^3^ in the head region.

### 2.3. Participant Posture

Participants were asked to lie supine in a relaxed state on a mat laid on a flat floor in a darkened room, which was maintained at an average temperature of 20 °C [15]. Each participant’s jaw position was relaxed without tooth contact to enable stabilization of the masticatory muscles. To reflect real-world resting conditions, participants were instructed to assume the most comfortable supine posture for each pillow height. They were able to maintain this self-selected position comfortably during the 5 min EMG recording sessions. This approach allowed for natural variation in head and neck posture, which was expected to influence OA and cervicohorizontal angles to some extent.

### 2.4. Measurement of OA

OA was defined as the angle formed between the maxillary occlusal plane and the horizontal floor surface in the supine position. The maxillary occlusal plane was identified using anatomical landmarks—the maxillary central incisal edges and the mesiobuccal cusps of the maxillary first molars. Standardized lateral photographs were taken for each participant under pillow height conditions of 0, 5, and 10 cm. The OA was then measured from the images using ImageJ software (Version 1.53t) (National. Institutes of Health, Bethesda, MD, USA) to ensure consistent angle analysis.

### 2.5. Measurement of MCM Activity Using Surface Electromyography Protocol

To record and analyze electromyography (EMG) signals from the MCM, we used the 10-channel Noraxon standard EMG system (MyoResearch XP Master Edition 1.08.27; Noraxon Inc., Scottsdale, AZ, USA). Surface electromyography (sEMG) muscle activity was detected using bipolar surface electrodes applied to the skin overlying the belly of each muscle, parallel to the muscle fibers. Recordings of the masseter (MAS) and temporalis (TEM) masticatory muscles on both sides and of the bilateral first cervical paravertebral muscle (PVM) and sternocleidomastoid muscle (SCM) cervical muscles were obtained. The MCM activities and OA were measured according to the changes in OA modulated by pillow heights. EMG signals were transmitted, amplified, and filtered (filtering bandpass, 20–350 Hz; rectification, smoothing root mean square, 50 ms). To allow normalization, participants performed maximum voluntary isometric contractions (MVCs) of each muscle prior to resting muscle activity measurements. The resting muscle activity ratios (RMRs) of each muscle were calculated by dividing the sEMG signal at rest by the sEMG signal during MVC. Resting muscle activities were recorded three times over a 5 min period for each OA modulated by pillow height and the mean values of the first 10 s of sEMG signals were calculated. After each measurement, participants were allowed to rest for 10 min. The measurement intervals at 0, 1, and 5 min were selected based on previous EMG studies demonstrating that muscle activity typically stabilizes within the first few minutes after adopting a new posture. This protocol enables the assessment of both immediate responses and short-term neuromuscular adaptation to pillow height conditions.

### 2.6. Statistical Analysis

The average resting muscle activity ratios (RMRs) were calculated for each OA modulated by pillow heights of 0, 5, and 10 cm at 0, 1, and 5 min after relaxation in the supine position. Normality of the data was assessed using the Shapiro–Wilk and Kolmogorov–Smirnov tests. Pearson correlation analysis was conducted to evaluate the relationships between OA and RMRs, as well as between OA and baseline characteristics such as age, height, weight, and body mass index (BMI). One-way and two-way analyses of variance (ANOVAs) were performed to compare differences in RMRs across time points and pillow height conditions. Effect sizes (η^2^) were also calculated to quantify the strength of association between factors and muscle activity. Given the repeated-measures nature of the design (multiple observations per subject across different time points and conditions), linear mixed-effects modeling or repeated-measures ANOVA could have offered more robust statistical inference. However, due to the limited sample size, only traditional ANOVA methods were applied in this study, and this limitation is acknowledged in the discussion. All statistical analyses were conducted using SPSS version 23.0 (IBM, Armonk, NY, USA), with significance set at *p* < 0.05. The intraclass correlation coefficient (ICC) was computed to evaluate the test–retest reliability of resting muscle activity measurements for the masseter (MAS) and sternocleidomastoid (SCM) muscles. A two-way mixed-effects model with absolute agreement (ICC [3, 1]) was used, and ICC values above 0.75 were interpreted as indicating good to excellent reliability [17].

## 3. Results

### 3.1. Association Between OA and Baseline Characteristics (Age, Height, Weight, and BMI) of Participants

The mean and standard deviation of the 23 participants’ age, height, weight, and BMI are presented in Table 1. Age and height showed a significant negative correlation with OA.

### 3.2. OA Change According to OA Modulated by Pillow Height

The OA values of the participants were measured at OA conditions modulated by pillow heights of 0, 5, and 10 cm. Even at the same OA modulated by pillow height, various OA values were observed (Table 1 and Figure 2) in the participants with varying weights and heights. The mean values of OA with 0, 5, and 10 cm pillows were 89.78 ± 8.67, 108.61 ± 7.36, and 121.0 ± 7.86, respectively (Figure 2).

### 3.3. RMR of MCMs According to Pillow Height

No significant correlation was found between RMRs of any MCMs and pillow height (*p* > 0.05; Figure 3). Although slight variations in RMR values were noted across different pillow heights, one-way ANOVA indicated no statistically significant differences (*p* > 0.05), suggesting that the observed variability does not represent consistent or systematic trends at the group level.

### 3.4. The RMR of MCM According to the OA

When OA was within 85°–104°, the RMR of the TEM, MAS, and PVM was the lowest, while within an OA of 116°–137°, the RMR of the SCM was the most reduced (Figure 4). The RMR of the TEM, MAS, and PVM increased significantly as OA increased (Figure 4). The effect sizes for MAS, TEM, PVM, and SCM are large, medium, small, and small, respectively (Table 2). The RMR of TEM and MAS increased significantly as the OA increased, and the RMR of the SCM decreased significantly as the OA increased (Figure 4).

K-means clustering on OA, MAS, and TEM RMR values identified two natural groupings of muscle activation levels. The elbow method (Appendix A) indicated k = 2 as the optimal number of clusters, revealing a low-activity group centered around 105°–111°, suggesting a potential comfort zone in supine positioning. These zones are visually summarized in Figure 5, which illustrates the defined comfort and risk regions based on clustering and EMG data.

### 3.5. Discriminative Ability of OA to Predict Muscle Hyperactivity

To evaluate the predictive capacity of the OA in identifying elevated resting muscle activity, receiver operating characteristic (ROC) analysis was performed. The OA demonstrated good discriminative performance for the MAS, with an area under the curve (AUC) of 0.83 and an optimal cut-off value of 138.0°, determined using Youden’s index. For the TEM, the AUC was 0.77, with a cut-off value of 133.1°, also based on Youden’s index.

These findings suggest that higher OA values may be associated with increased risk of masticatory muscle hyperactivity in the supine position (Figure 6).

### 3.6. Linear Regression Analysis of OA and Muscle Activity

To investigate the relationship between OA and the normalized activity of masticatory muscles, ordinary least squares (OLS) regression was performed using pooled data across the 0, 1, and 5 min time points. In the case of the MAS, OA was found to be a significant positive predictor of muscle activity. The regression model demonstrated a clear upward trend in MAS RMR with increasing OA (Figure 7A), suggesting that a larger occlusal angle may be associated with elevated masticatory muscle activation in the supine position. For the TEM, a similar trend was observed (Figure 7B). Although the magnitude of the increase was less steep than for MAS, the regression line indicated a consistent linear association between OA and TEM RMR across all time points.

In both muscles, the observed scatterplot data were dispersed around the regression line but maintained a visually interpretable directional trend. These findings support the hypothesis that changes in OA, possibly induced by pillow height, influence the tonic activation of jaw muscles even during short-term supine positioning. However, since ordinary least squares regression does not account for intra-subject variability or repeated measures, these findings represent population-level tendencies. Detailed coefficients and significance levels for each muscle-specific linear mixed-effects model, including interaction terms, are provided in Appendix A.

### 3.7. Linear Mixed-Effects Model Analysis of RMR According to OA and Time

To account for within-subject variability and repeated measurements, linear mixed-effects models (LMMs) were used to assess the effects of OA, time (0, 1, 5 min), and their interaction on RMRs across four muscles: MAS, TEM, SCM, and PVM.

The results revealed that OA was a significant positive predictor of RMR in MAS (β = 0.0034, *p* = 0.005) and TEM (β ≈ 0.0022, *p* < 0.001), suggesting increased masticatory muscle activity with greater OA. In contrast, SCM exhibited a significant negative association with OA (β = –0.0026, *p* = 0.001), indicating reduced activity at higher OA values. The effect of OA on PVM was not statistically significant (β = –0.0007, *p* = 0.12).

Importantly, a significant interaction between OA and time (OA × time) was observed for the masseter muscle (β = –0.0013, *p* = 0.017), indicating that the effect of OA on MAS activity varied depending on the time point (0, 1, or 5 min). No significant interaction effects were observed for TEM, SCM, or PVM (Appendix A). These findings are visually summarized in Figure 8, which presents the beta coefficients from the LMM analysis for each muscle.

These findings support the hypothesis that OA modulates resting muscle activation in a muscle-specific and time-dependent manner, emphasizing the importance of individualized occlusal posture in ergonomic or therapeutic applications.

## 4. Discussion

Forward head position (FHP), which is defined as a sitting posture with the ear tragus positioned forwards from the shoulder or trunk, was found to be a contributing factor to TMD, and the greater the forward head posture, the greater the MAS activity [18,19,20]. These results have been used as a basis for behavioral therapy in TMD. Similarly, FHP patients complain of pain in the MCM after sleep or an activity in lying position [21,22,23]. Data regarding the posture with lowered RMR of MCM are needed, especially those pertaining to masticatory muscles [13,24,25,26].

The resting activities of MAS and TEM, which are directly attached to the mandible, may change with angle changes in the head axis with respect to the ground in the lying posture. The head axis is almost perpendicular to the maxillary occlusal surface, and the angle of the head axis with respect to the ground has a proportional relationship with OA. In the lying posture, the mandible is affected by gravity. The effect of gravity on the mandible can be increased as the axis of the head is raised against the ground in the lying posture [27]. Thus, if the gravity applied to the mandible changes, the muscle activities of the MAS and TEM attached to the mandible may change [12]. This interpretation is supported by the linear regression models (Figure 7), which demonstrated a significant positive association between OA and both MAS and TEM RMRs. Notably, the OLS-derived regression lines indicated a stronger slope for MAS than for TEM, suggesting that OA changes have a more pronounced influence on MAS activation than on TEM.

While the OA values themselves showed a significant association with RMR, the pillow height did not show a statistically significant difference (*p* > 0.05), suggesting that individual anatomical factors may have stronger influence than pillow height per se. These findings suggest that OA should be considered to help avoid unnecessary MCM activities in supine position.

The perception of comfort is associated with the low muscle activity resulting from better alignment of the head, neck, and shoulders, which helps maintain a proper muscle length to create isometric tension and balance [28]. According to our results, the application of OA adjusted to 85°–104° could minimize unnecessary tension of the MAS, TEM, and PVM, and might reduce disease-related pain and deterioration in individuals who are vulnerable to TMDs. On the other hand, SCM has the lowest muscle activity at 116°–137°, which could also minimize the unnecessary tension in SCM-related problems. The resting activities of MAS were greatly influenced by the changes in OA, followed by TEM, PVM, and SCM (Table 2). Additionally, k-means clustering analysis of OA, MAS, and TEM activity revealed two distinct clusters, with one centered around 105–111°, indicating a low-muscle-activation “comfort zone.” The optimal number of clusters (k = 2) was determined using the elbow method, which identified a clear point of inflection in the WCSS plot. This clustering supports the presence of physiologically distinct postural groups: one with minimal masticatory muscle activity and another with elevated activation. Appendix A illustrates this cluster separation. This result, along with the OLS regression findings in Figure 7, supports the interpretation that MAS is the most OA-sensitive muscle among those analyzed. The observed-to-predicted patterns also confirm that OA can serve as a continuous predictor of masticatory muscle load in the supine posture. Bland–Altman plots confirmed the consistency of repeated EMG measurements within ± 1.96 SD. In the Bland–Altman plots, the 95% limits of agreement for MAS were −4.21% to +5.03%, and for SCM were −3.79% to +4.18%, indicating good repeatability with minimal measurement bias (Appendix A). ICC for repeated RMR measurements showed excellent reliability for both muscles, MAS (ICC2 = 0.82) and SCM (ICC2 = 0.79), indicating strong test–retest consistency across measurement points. While the OA range of 85°–104° was associated with the lowest muscle activation across MAS, TEM, and PVM—suggesting potential for minimizing functional strain—clustering analysis further refined this finding by identifying 105°–111° as the statistical center of a low-activity group. This range may thus represent a more practical ‘comfort zone’ for most individuals, where relaxation across all target muscles and biomechanical feasibility are balanced.

Although the ROC analysis identified 138° as the optimal cut-off for predicting high masseter activity, the k-means clustering suggested a natural low-activity posture zone between 105° and 111°. This discrepancy likely reflects the methodological difference in focus: ROC identifies a threshold to separate risk, while clustering detects the region with inherently minimal muscle activity. Therefore, the OA range of 105°–111° may serve as an optimal comfort zone, whereas OA above 138° may indicate potential muscular overload, especially in vulnerable individuals. These dual criteria can together guide both preventive screening and personalized pillow adjustment. The ROC-derived cut-off (138°) represents a risk threshold for excessive muscle activation, while the 105°–111° range from k-means clustering reflects a comfort zone of minimal EMG activity. This distinction highlights the dual application of OA in both risk screening and ergonomic optimization.

Despite the insights provided, several limitations of the present study should be acknowledged. First, the sample size of 23 participants limits the statistical power and generalizability of the results. Although repeated measures were obtained, mixed-effects modeling—commonly used for within-subject variability—was not employed, which may limit the robustness of the observed trends. Second, the OA was measured relative to the ground rather than using standardized lateral cephalometric imaging. This may raise concerns about measurement reproducibility and clinical applicability in diverse settings. Third, muscle activity was recorded over a short 5 min supine session, without consideration of longer-term or sleep-associated changes. Given that muscle activity may fluctuate over time during sleep, short-term recordings may not fully reflect real-world resting conditions. Fourth, although RMRs were compared across multiple muscles, intrinsic anatomical and functional differences were not fully accounted for in the analysis. For example, the SCM has a wide and functionally diverse structure, complicating straightforward interpretation of its RMR values. Future studies incorporating longer-term observations, cephalometric validation, and more advanced statistical modeling (e.g., linear mixed-effects models) are warranted to strengthen the reliability and clinical relevance of the findings. In addition, the results should be interpreted with caution and validated in studies involving larger populations and real-world sleep conditions.

## 5. Conclusions

Our study findings indicate that OA appears to be more closely related with resting MCM activities than pillow height. As the resting activities of MAS and TEM had a stronger correlation with OA than SCM and PVM, determining optimal OA by adjusting OA should be considered to minimize unnecessary resting activities of MAS and TEM. Reducing MCM activities results in reduced pain and further affects the quality of sleep. The OA thresholds identified in this study are proposed as ergonomic guidance ranges, not as absolute diagnostic cut-offs.

## Figures and Tables

**Figure 1 medicina-61-01274-f001:**
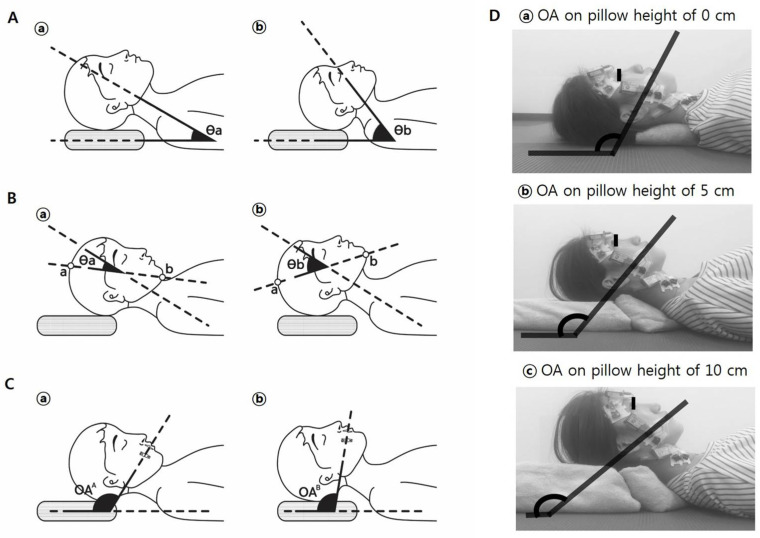
(**A**) Cervicohorizontal angles can vary despite using the same pillow height. (**a**) A large individual; (**b**) a small individual. Angle θa may be smaller than θb. (**B**) Craniocervical angles also differ despite the same pillow height. The individuals in (**a**,**b**) have similar physical dimensions, but θa differs from θb. a, vertex; b, mentum. (**C**) Occlusal angles (OAs) can also vary under the same pillow height. OA is defined as the angle between the maxillary occlusal plane and the ground. The individuals in (**a**,**b**) are of similar size, yet OA^A^ differs from OA^B^. (**D**) Representative photos showing that the self-selected comfortable posture, including the OA, varied depending on pillow height.

**Figure 2 medicina-61-01274-f002:**
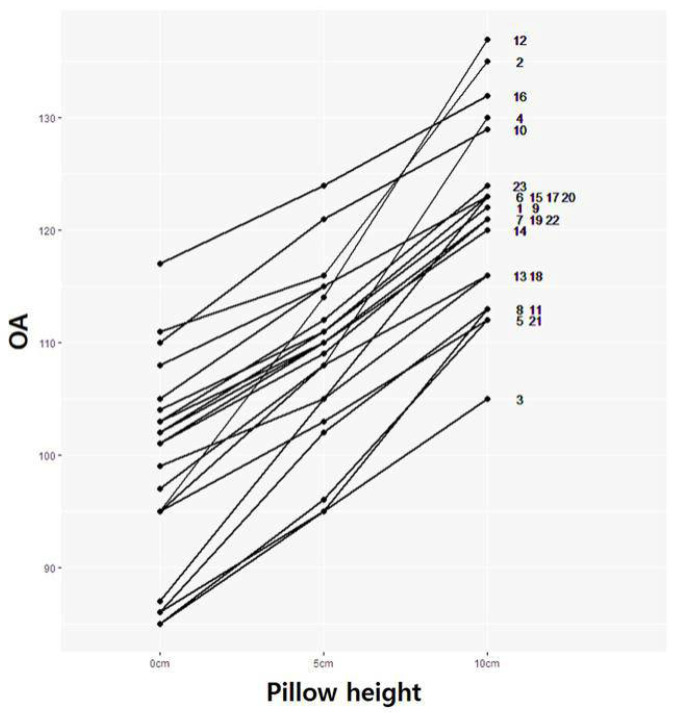
Various degrees of individual occlusal angle changes according to pillow height in participants 1–23. OA, occlusal angle; 1–23, participant’s number.

**Figure 3 medicina-61-01274-f003:**
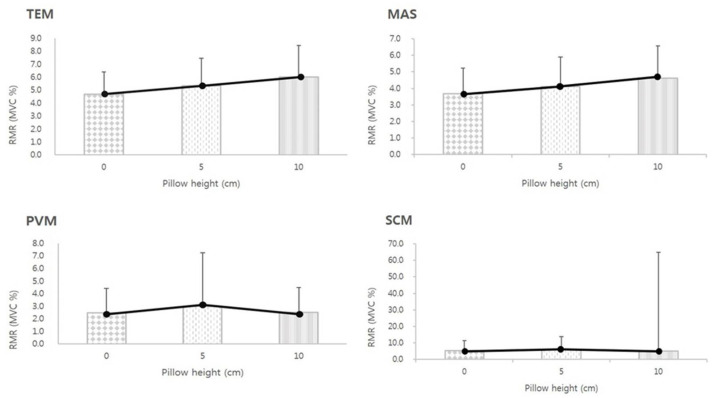
Analysis of the resting muscle activity ratio according to pillow height in the supine position (*p* > 0.05, one-way analysis of variance). Error bars represent standard deviations. TEM, temporalis; MAS, masseter; PVM, first cervical paravertebral muscle; SCM, sternocleidomastoid; RMR (MVC%), ratio of resting muscle activity/maximal muscle contraction; MVC, maximum voluntary isometric contraction.

**Figure 4 medicina-61-01274-f004:**
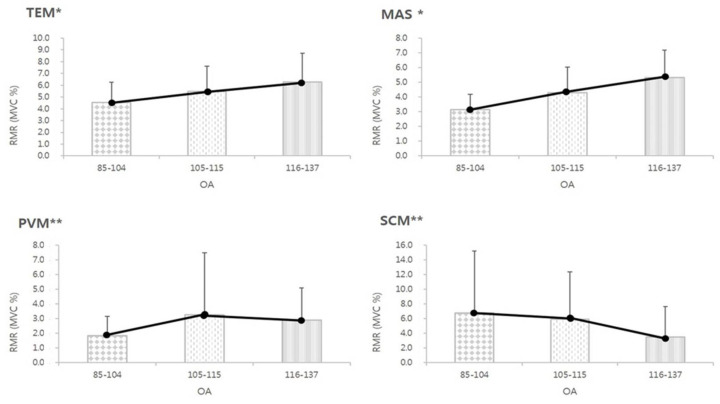
Resting muscle activity ratio according to occlusal angle in the supine position (one-way analysis of variance). Error bars represent standard deviations. OA, occlusal angle; TEM, temporalis; MAS, masseter; PVM, first cervical paravertebral muscle; SCM, sternocleidomastoid; RMR (MVC%), resting muscle activity ratio/maximal muscle contraction; MVC, maximum voluntary isometric contractions. * *p* < 0.001, ** *p* < 0.05.

**Figure 5 medicina-61-01274-f005:**
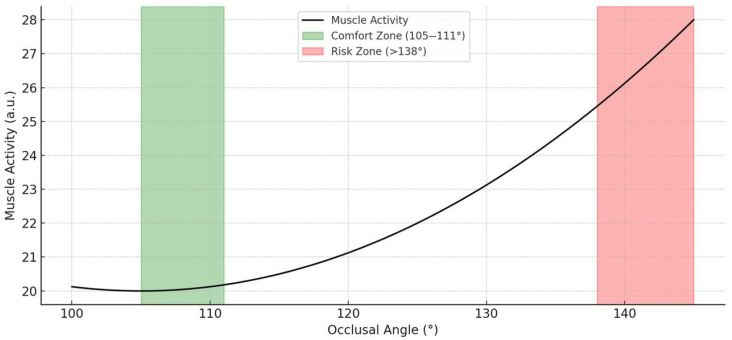
Graphic representation of occlusal angle comfort and risk zones. The graph illustrates the relationship between occlusal angle (OA) and estimated muscle activity levels. The green-shaded region (105–111°) represents the defined comfort zone, where masticatory and cervical muscle activity is minimized. The red-shaded region (>138°) indicates the risk zone, associated with increased neuromuscular activation. These zones were determined based on observed EMG trends and clustering analysis.

**Figure 6 medicina-61-01274-f006:**
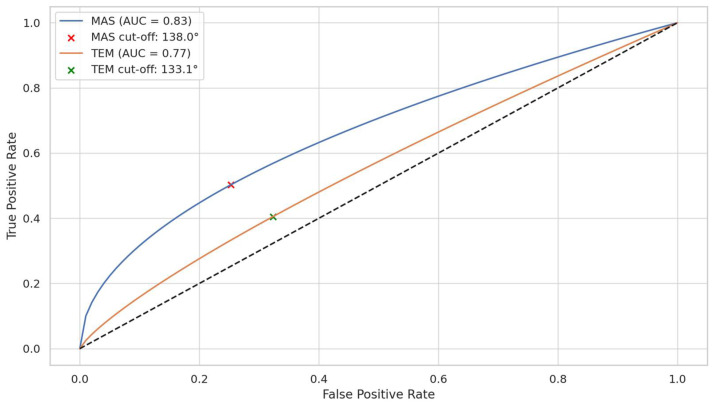
Receiver operating characteristic (ROC) curves of occlusal angle (OA) for predicting hyperactivity in the masseter (MAS) and temporalis (TEM) muscles. The ROC analysis demonstrates the discriminative performance of OA in identifying elevated resting muscle activity. The MAS curve (blue) shows an area under the curve (AUC) of 0.83 with an optimal cut-off value of 138°, while the TEM curve (green) demonstrates an AUC of 0.77 with a cut-off value of 133.1°. Red and green cross mark indicate the respective optimal thresholds calculated using Youden’s index. The black dashed line represents the reference line for a random classifier (AUC = 0.5). These results suggest that OA is a more reliable predictor of hyperactivity in MAS than in TEM.

**Figure 7 medicina-61-01274-f007:**
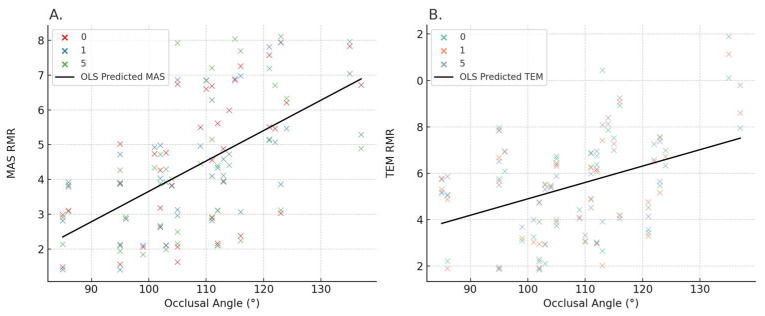
Predicted resting muscle activity (RMR) across occlusal angle (OA) for MAS, TEM, SCM, and PVM based on linear mixed-effects modeling (LMM). The figure illustrates the modeled association between OA and RMR values for four muscles. A significant positive relationship was observed for MAS and TEM, indicating increased masticatory muscle activity with larger OA values. In contrast, a significant negative association was found for SCM, while the trend for PVM was not statistically significant. The plots represent marginal means across three time points (0, 1, and 5 min), capturing temporal variation. Notably, a significant OA × time interaction was found in MAS, highlighting time-dependent modulation of masticatory muscle activity with changing OA. (**A**) *MAS*: A significant positive linear relationship was observed between OA and RMR, with increasing OA associated with elevated masseter muscle activity. (**B**) *TEM*: A similar positive association was found between OA and temporalis RMR, although the slope was less steep than for MAS. The regression lines represent OLS-predicted values.

**Figure 8 medicina-61-01274-f008:**
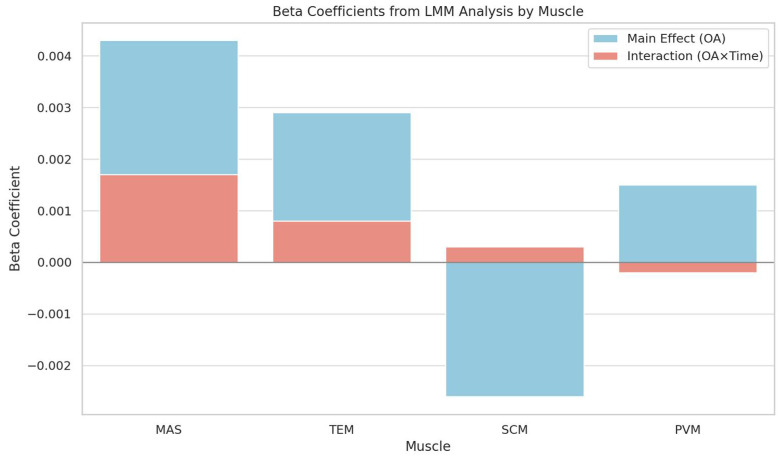
Beta coefficients from linear mixed-effects models (LMMs) showing the main effect of occlusal angle (OA) and the interaction between OA and time (OA × time) on resting muscle activity (RMR) across four muscles: masseter (MAS), temporalis (TEM), sternocleidomastoid (SCM), and paravertebral muscles (PVM). The blue bars represent the fixed effect of OA; red bars represent the OA × time interaction term.

**Table 1 medicina-61-01274-t001:** Association between occlusal angle (OA) and baseline characteristics (age, height, weight, and BMI) of participants.

	Male	Female	Total	r
Age (year)	36 ± 13.53	32.59 ± 11.50	42.44 ± 15.36	−0.28 *
Height (cm)	165.42 ± 9.2	160.82 ± 6.93	174.11 ± 6.21	−0.35 **
Weight (kg)	62.1 ± 17.56	52.97 ± 6.79	79.33 ± 18.98	−0.25
Body Mass Index (kg/m^2^)	22.32 ± 3.91	20.41 ± 1.40	25.93 ± 4.62	−0.18
OA (5 cm) ^1^	104.67 ± 9.98	110.44 ± 5.43	108.86 ± 7.43	

^1^ OA (5 cm): occlusal angles of each participant which were measured when using a 5 cm pillow; r: Pearson correlation coefficient according to occlusal angle (OA); * *p* < 0.05, ** *p* < 0.01.

**Table 2 medicina-61-01274-t002:** Effect sizes for masticatory and cervical muscle activities according to occlusal angle ^†^.

	OA		
	F-Stat	*η* ^2^	Effect Size
TEM	9.591 **	0.108	Medium
MAS	25.437 **	0.242	Large
PVM	3.609 *	0.043	Small
SCM	3.460 *	0.042	Small

OA, occlusal angle; TEM, temporalis; MAS, masseter; PVM, first cervical paravertebral muscle; SCM, sternocleidomastoid; RMR (EMG ratio), resting muscle activity ratio/maximal muscle contraction; MVC, maximum voluntary isometric contractions. ^†^ One-way analysis of variance. * *p* < 0.05, ** *p* < 0.001. Grading of effect size of eta-squared20: 0.01 ≤ small < 0.06; 0.06 ≤ medium < 0.14; 0.14 ≤ large.

## Data Availability

All data relevant to the study are included in the article.

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
