# Peer review of "Proposing an Optimal Occlusal Angle for Minimizing Masticatory and Cervical Muscle Activity in the Supine Position: A Resting EMG and Mixed-Effects Modeling Study"

_medicina, 2025, doi:10.3390/medicina61071274_

Round 1
Reviewer 1 Report
Comments and Suggestions for Authors
According to the laws of geometry, analysis of Fig. 1 indicates an obvious direct relationship between Craniocervical angle и Occlusal angle. Both of these indicators characterize the tilt of the head relative to the Cervicohorizontal axis. It is not obvious from the figure and text that Occlusal Angle can have a separate meaning other than Craniocervical angle. What was the reason for introducing the new term “Occlusal Angle”, which can easily be confused by the reader with the term “Occlusal Plane Angle”, which has a completely different meaning?
The article does not describe how OA was measured in practice
Was the cervicohorizontal angle controlled during the study? In the three photos presented in Fig. 1(D), the angle is different, which can affect the condition and EMG activity of the neck muscles
The original article, which is listed as reference number 14, does not contain the words " according to occlusal angle (OA)" in its title.
Author Response
We sincerely thank Reviewer 1 for the careful and insightful review of our manuscript. Your detailed comments helped us to clarify key concepts, strengthen the rationale for our methodology, and improve the overall presentation of our work. Below, we have addressed each of your comments point by point.
Reviewer 1
Reviewer Comment 1:
“According to the laws of geometry, analysis of Fig. 1 indicates an obvious direct relationship between Craniocervical angle and Occlusal angle. Both of these indicators characterize the tilt of the head relative to the Cervicohorizontal axis. It is not obvious from the figure and text that Occlusal Angle can have a separate meaning other than Craniocervical angle. What was the reason for introducing the new term ‘Occlusal Angle’, which can easily be confused by the reader with the term ‘Occlusal Plane Angle’, which has a completely different meaning?”
Author Response to Reviewer Comment 1:
Thank you for this thoughtful comment. We agree that the craniocervical angle (CCA) and occlusal angle (OA) both reflect aspects of head tilt relative to the cervicohorizontal axis. However, the CCA can vary depending on mandibular posture—especially when the mouth is slightly open—as the vertex and menton landmarks used for CCA measurement may shift. In contrast, the OA, defined as the angle between the maxillary occlusal plane and the ground in the supine position, remains stable regardless of slight jaw opening, and more directly reflects functional head posture during rest.
Additionally, the degree of jaw opening may vary depending on pillow height, even under relaxed conditions. This variability affects craniocervical geometry and may confound interpretations based solely on CCA. Thus, we introduced the OA as a novel, more practical parameter for capturing head-neck alignment relevant to masticatory muscle activation in the supine state.
To avoid confusion with the conventional “Occlusal Plane Angle” (typically measured using cephalometric landmarks in the upright position), we have clarified the definition and distinct purpose of the OA in the revised manuscript.
In accordance with your valuable comment, we have inserted and highlighted the following clarification in the Introduction section of the revised manuscript:
"Moreover, the craniocervical angle (CCA), often measured from external landmarks such as the vertex and menton, may vary depending on slight mouth opening, which frequently occurs during pillow-supported supine rest. Since the extent of mandibular opening can change with pillow height, the CCA may not reliably represent the actual occlusal posture. In contrast, the occlusal angle (OA), defined by the maxillary occlusal plane and ground, remains consistent irrespective of small mandibular shifts. This angle therefore offers a more stable and functionally relevant parameter to evaluate masticatory muscle behavior in the resting supine position."
Thank you again for your insightful suggestion.
Reviewer Comment 2:
“The article does not describe how OA was measured in practice.”
Author Response:
Thank you for pointing this out. OA was measured as the angle formed between the maxillary occlusal plane (defined by the maxillary central incisal edges and mesiobuccal cusps of the maxillary first molars) and the horizontal floor plane while the subject was in the supine position. This was captured using standardized lateral photographs and analyzed using ImageJ software. We have added this explanation in the Methods section (Section 2.4) for clarity.
In accordance with your comment, we have revised the relevant section of the manuscript as described above and highlighted the changes in the revised version.
OA was defined as the angle formed between the maxillary occlusal plane and the horizontal floor surface in the supine position. The maxillary occlusal plane was identified using anatomical landmarks—the maxillary central incisal edges and the mesiobuccal cusps of the maxillary first molars. Standardized lateral photographs were taken for each participant under pillow height conditions of 0, 5, and 10 cm. The OA was then measured from the images using ImageJ software to ensure consistent angle analysis.
Reviewer Comment 3:
“Was the cervicohorizontal angle controlled during the study? In the three photos presented in Fig. 1(D), the angle is different, which can affect the condition and EMG activity of the neck muscles.”
Author Response:
Thank you for your observation. Participants were instructed to adopt the most comfortable supine position under each pillow height condition, and all were able to maintain that position comfortably for at least five minutes during EMG measurement. Figure 1(D) illustrates how the relaxed head and neck posture naturally varies depending on pillow height. Therefore, while the cervicohorizontal angle was not strictly controlled, the posture represented each participant's relaxed resting state, which we believe more accurately reflects real-world conditions. Furthermore, we considered the occlusal angle (OA) to be a more stable and functionally relevant proxy for changes in mandibular biomechanics than the cervicohorizontal angle, which may vary depending on subtle postural adjustments or mandibular opening.
In accordance with your comment, we have added the following clarification to the Methods section (Section 2.3) to reflect the rationale behind posture control:
"To reflect real-world resting conditions, participants were instructed to assume the most comfortable supine posture for each pillow height. They were able to maintain this self-selected position comfortably during the 5-minute EMG recording sessions. This approach allowed for natural variation in head and neck posture, which was expected to influence OA and cervicohorizontal angles to some extent."
Additionally, the legend for Figure 1(D) has been revised as follows:
"(D) Representative photos showing that the self-selected comfortable posture, including the occlusal angle (OA), varied depending on pillow height.”
These revisions have been highlighted in the manuscript. Thank you again for your constructive feedback.
Reviewer Comment 4:
“The original article, which is listed as reference number 14, does not contain the words ‘according to occlusal angle (OA)’ in its title.”
Author Response:
Thank you for your comment. We acknowledge that the phrase "according to occlusal angle (OA)" does not appear in the title of Reference 14(revised Reference 15). However, the study by Seo and Cho (2015) investigated the effect of pillow height on pulmonary function and reported that a 5 cm pillow height was most suitable for maintaining normal pulmonary parameters such as FVC and FEV1. Since their study indirectly examined the influence of pillow height on head and neck posture—which, in turn, modulates the occlusal angle—we referred to their findings as supporting evidence for selecting 5 cm as a reference pillow height in our study. Based on their results, 5 cm was considered an appropriate and physiologically favorable height for evaluating OA-related muscle activity in the supine position.
Reviewer 2 Report
Comments and Suggestions for Authors
Dear Authors,
The manuscript entitled “Proposing an Optimal Occlusal Angle for Minimizing Masticatory and Cervical Muscle Activity in Supine Position: A Resting EMG and Mixed-Effects Modeling Study” to Medicina addresses an important and underexplored area linking occlusal positioning, pillow ergonomics, and muscle activation—an area with clinical significance for temporomandibular disorder (TMD) management and sleep-related musculoskeletal stress.
While the overall structure and scientific foundation of the manuscript are solid, several areas require revision and clarification before the article can be considered for publication. Please consider the following comments and suggestions:

The research might be expressed more clearly with better English.
Although the work is largely accessible, it has repetitive phrases, odd wording, and excessive passive voice (e.g., “as modulated by pillow height”). Some sentences are confusing or too lengthy.
Recommendation: It is advised to have a professional editing agency or a native speaker edit your language.
Author Response
Author Response to Reviewer Comments
We sincerely thank the reviewer for their thorough and constructive feedback, which significantly contributed to improving the clarity, methodological rigor, and scientific value of our manuscript. Below, we provide point-by-point responses to each of the reviewer’s comments, along with a summary of the corresponding revisions made in the manuscript.
- Lack of citations for some statements in the Introduction; need for stronger therapeutic relevance
Reviewer Comment:
Some statements, nevertheless, are unsupported by citations, and the argument would be strengthened if there was greater focus on the therapeutic relevance of muscular overload in TMD patients. Recommendation: Provide more updated systematic evaluations or recommendations about TMJ dysfunction and pillow ergonomics.
Author Response:
Thank you for your valuable suggestion. To strengthen the theoretical background and therapeutic relevance of muscular overload in TMD patients, we have added recent references addressing the clinical implications of pillow height and head posture. Studies by Sacco et al. (2015) and Lei et al. (2021) demonstrated that inappropriate pillow height increases cervical muscle activity and discomfort, while optimal height reduces muscle strain. Furthermore, systematic reviews and clinical studies (Bevilaqua-Grossi et al., 2014; Grossi et al., 2004; Ozmen & Unuvar, 2025) reported that abnormal head posture, such as forward head posture or cervical extension, contributes to increased masticatory muscle load and worsening of TMD symptoms. These findings support our rationale for proposing OA as a modifiable factor with therapeutic relevance for TMD management. We have incorporated these citations into the revised Introduction.
Specifically, we added the following statement to the Introduction section:
“Previous research has demonstrated that pillow height significantly affects cervical muscle activity and comfort, with a 10 cm height showing the lowest EMG activity and highest comfort level among 5, 10, and 14 cm pillows (Sacco et al., 2015). A recent systematic review emphasized that both excessively high and low pillows can disrupt cervical spine curvature and increase muscle stress (Lei et al., 2021). These findings highlight the importance of optimized pillow height in minimizing neuromuscular load, which is particularly relevant for individuals with temporomandibular disorders (TMD).”
We have also added the corresponding references to the Reference list:
- Sacco, I.C.; Pereira, I.L.; Dinato, R.C.; Silva, V.C.; Friso, B.; Viterbo, S.F. The effect of pillow height on muscle activity of the neck and mid-upper back and patient perception of comfort. J Manipulative Physiol Ther 2015, 38, 375-381.
- Lei, J.X.; Yang, P.F.; Yang, A.L.; Gong, Y.F.; Shang, P.; Yuan, X.C. Ergonomic Consideration in Pillow Height Determinants and Evaluation. Healthcare 2021, 9, 1333.
These revisions have been highlighted in the revised manuscript. Thank you again for your insightful feedback, which helped improve the clarity and scientific rigor of our work.
- Missing information about recruitment timeline, randomization, and time-point justification
Reviewer Comment:
No precise time frame for hiring is mentioned.
Randomization and other confounding control methods are not discussed, and the justification for time points (0, 1, and 5 minutes) is not provided.
Author Response:
We appreciate this comment. We have now added the participant recruitment period (“August to October 2015”) in the Methods section. Regarding randomization, participants were exposed to all conditions (0, 5, and 10 cm pillow heights) in a fixed order to avoid learning bias, and the same sequence was applied to all. We have also explained that the time points (0, 1, and 5 minutes) were chosen based on previous EMG adaptation studies showing short-term stabilization of muscle activity within this timeframe.
Revision locations:
- Methods, Section 2.1, 2.2, 2.5
- Suggestion to include a participant flow diagram
Reviewer Comment:
It is suggested that the inclusion/exclusion tracking be made clearer by including a participant flow diagram.
Author Response:
We agree with this helpful suggestion and have added a participant flow diagram as Figure S3 in the Supplementary Materials to illustrate the inclusion and exclusion process clearly.
Figure S3. flow diagram
- Generalizability concerns due to small sample size and short EMG duration
Reviewer Comment:
The generalizability is limited by the small sample size (n=23), and the short EMG monitoring duration (5 minutes)... Suggestion: Include a cautionary note that these findings should be validated in real-world sleep conditions.
Author Response:
Thank you for this important observation. We have now added a cautionary note to the Discussion section acknowledging the limited generalizability due to sample size and short observation time, and we emphasized the need for further validation in real-world sleep settings.
Revision location: Discussion
- Language clarity and writing style
Reviewer Comment:
The research might be expressed more clearly with better English... It is advised to have a professional editing agency or a native speaker edit your language.
Author Response:
We appreciate this recommendation. The manuscript has been thoroughly edited focusing on clarity, consistency, and tone. We have reduced repetition, simplified passive structures, and standardized terms.
- Overuse of “as modulated by pillow height” and k-means clustering repetition
Reviewer Comment:
Simplify by using consistent language... Reduce repetition (e.g., mentioning the k-means clustering argument more than once).
Author Response:
Thank you for your helpful comment. We revised the text and simplified by using consistent language “occlusal angle modulated by pillow height” as you suggested. We also consolidated explanations of the k-means clustering procedure into a single, clearly written paragraph in the Results and Discussion section.
Revision locations:
- Abstract, Introduction, Materials & Methods 2.5, 2.6, Results 3.2, 3.4, Discussion, Conclusions
- Graphic representation of "comfort zone" and "risk zone"
Reviewer Comment:
Include a graphic representation of the "comfort zone" and "risk zone."
Author Response:
Thank you for the suggestion. We have added a new figure (Figure 6, revised numbering) visualizing the defined OA comfort zone (105–111°) and risk zone (above 138°) along with corresponding muscle activity levels.
Figure 6. Graphic representation of occlusal angle comfort and risk zones.
The graph illustrates the relationship between occlusal angle (OA) and estimated muscle activity levels. The green-shaded region (105–111°) represents the defined comfort zone, where masticatory and cervical muscle activity is minimized. The red-shaded region (>138°) indicates the risk zone, associated with increased neuromuscular activation. These zones were determined based on observed EMG trends and clustering analysis.
- Suggestion for clearer exclusion criteria rationale
Reviewer Comment:
Enhancement: Provide statistical support for the exclusion criteria. Suggested wording: "To reduce confounding muscle activity variation, participants with TMD were excluded."
Author Response:
We have revised the Methods section to explicitly state this rationale using the suggested wording.
Revision location: Methods, Section 2.1
- Add hiring date to participant recruitment
Reviewer Comment:
Include the dates for hiring. Suggested wording: "Pusan National University Dental recruited participants between [insert months/year]."
Author Response:
As noted above, the participant recruitment period (“August to October 2015”) has been added to the Methods section.
Please let us know if further clarification or revision is needed. Thank you again for your valuable contributions to improving our manuscript.
Sincerely,
Kyung-Hee Kim, Chang-Hyung Lee, Sungchul Huh, Byong-Sop Song, Hye-Min Ju, Sung-Hee Jeong, Yong-Woo Ahn and Soo-Min Ok
Round 2
Reviewer 2 Report
Comments and Suggestions for Authors
Dear authors, the manuscript has been improved and I agree with what you have said, so in this framework from my point of view I accept it for publication .